# Hearing Loss Caused by HCMV Infection through Regulating the Wnt and Notch Signaling Pathways

**DOI:** 10.3390/v13040623

**Published:** 2021-04-06

**Authors:** Sheng-Nan Huang, Yue-Peng Zhou, Xuan Jiang, Bo Yang, Han Cheng, Min-Hua Luo

**Affiliations:** 1State Key Laboratory of Virology, CAS Center for Excellence in Brain Science and Intelligence Technology Wuhan Institute of Virology, Wuhan 430071, China; ShengnanHuang_wh@outlook.com (S.-N.H.); pengchengwanli1994@outlook.com (Y.-P.Z.); jiangx@wh.iov.cn (X.J.); michealisyang@gmail.com (B.Y.); 2University of Chinese Academy of Sciences, Beijing 100049, China; 3Joint Center of Translational Precision Medicine, Guangzhou Institute of Pediatrics, Guangzhou Women and Children Medical Center, Guangzhou 510000, China; 4Shanghai Public Health Clinical Center, Fudan University, Shanghai 201508, China

**Keywords:** cytomegalovirus, congenital cytomegalovirus infection, sensorineural hearing loss, inner ear development, Wnt signaling pathway, Notch signaling pathway

## Abstract

Hearing loss is one of the most prevalent sensory disabilities worldwide with huge social and economic burdens. The leading cause of sensorineural hearing loss (SNHL) in children is congenital cytomegalovirus (CMV) infection. Though the implementation of universal screening and early intervention such as antiviral or anti-inflammatory ameliorate the severity of CMV-associated diseases, direct and targeted therapeutics is still seriously lacking. The major hurdle for it is that the mechanism of CMV induced SNHL has not yet been well understood. In this review, we focus on the impact of CMV infection on the key players in inner ear development including the Wnt and Notch signaling pathways. Investigations on these interactions may gain new insights into viral pathogenesis and reveal novel targets for therapy.

## 1. Introduction

Disabling hearing loss is defined by the World Health Organization (WHO) as a permanent hearing loss, in the better hearing ear of greater than 40 decibels (dB) in adults and 30 dB in children. It affects 466 million people including 34 million children worldwide. Most of the affected ones live in low- and middle-income countries with limited ear and hearing care services. Particularly for children, disabling hearing loss has devastating consequences because it seriously hinders them from developing language, communication and social skills with inferior long-term education and economic status. The leading nongenetic cause of sensorineural hearing loss (SNHL) in children is congenital human cytomegalovirus (HCMV) infection, which affects 0.5–3% of all live births worldwide [1]. Up to 60% of symptomatic infants with HCMV infection will suffer from permanent sequelae, with SNHL being the most common abnormality. While asymptomatic infants are at a lower risk of progression to severe illness, 10–15% of them will subsequently develop SNHL. Thus, the case number of SNHL from asymptomatic children is actually larger than that from symptomatic newborns [1].

HCMV is a member of subfamily β-herpesvirinae in the family Herpesviridae. Primary HCMV infection typically causes no symptoms, but HCMV establishes lifelong latent infection, which can be periodically reactivated with shedding of infectious virus. Women of reproductive age are predominantly affected by HCMV infection, with seroprevalence ranging from 45% to 100% [2]. Primary or recurrent HCMV infection during pregnancy can lead to vertical transmission to the fetus, which causes congenital infection with rates of 32.3% and 1.4% respectively [3]. Congenital HCMV infection is one of the most common causes of birth defects and childhood disability, with a higher prevalence than a few well-known pediatric conditions including Down syndrome, spina bifida, and fetal alcohol spectrum disorders [4].

Currently, no vaccine is available to prevent primary or recurrent HCMV infection and medical treatment options are also very limited. A thorough understanding of how HCMV infection causes hearing loss will help to identify novel targets for therapeutic intervention. In this review, we first give a short overview of hearing loss and then discuss a few potential routes that CMV hijacks to disrupt the inner ear development. The role of connexin 43 in hearing loss and its connection to CMV infection are discussed. More importantly, two essential signaling pathways-Notch and Wnt pathways and their interplay with CMV are explored.

## 2. Hearing Loss and HCMV Infection

In mammals, the ear can be divided into three parts: the outer ear, the middle ear and the inner ear. The inner ear is located within the temporal bone and lies between the middle ear and the internal acoustic meatus. It consists of three parts: the semicircular canals, the vestibular and the cochlear. The cochlea with a snail-shaped structure receives incoming soundwaves and transduces them into nerve impulses which are conveyed to the auditory nuclei in the brain where the signals are interpreted as sound. The transduction of auditory signals is carried out in the organ of Corti which sits upon the vestibular surface of the basilar membrane within the cochlear duct. Inside the organ of Corti, a single row of inner hair cells (IHCs) and three rows of outer hair cells (OHCs) are arranged along rows and are interspaced by the supporting cells (SCs) [5]. The receptors on the surface of hair cells (HCs) can sensor the shifts between the tectorial and basilar membranes, resulting in release of glutamate from the HCs onto the auditory nerve which sends nerve impulses to the brain (Figure 1).

There are two main types of hearing loss based on the structure and function [6]. First, sensorineural hearing loss (SNHL) is the most common type of hearing loss and usually not medically or surgically treatable. It occurs when the inner ear nerves and HCs are damaged, possibly by aging, noise exposure, heredity, or diseases. Second, conductive hearing loss happens when the transmission of sound waves is blocked in the outer or middle ear, typically caused by earwax buildup, fluid, or a punctured eardrum. This type of hearing loss is often temporary and can be restored by medical treatment. 

Moreover, hearing loss can be categorized as congenital (present at birth) or acquired hearing loss (after birth) based on the timing that hearing loss appears. Almost 50% of all congenital SNHL cases are hereditary and caused by mutations in genes such as GJB2, GJB6, SLC26A4, and OTOF [7]. Viral infection is another major risk factor for hearing loss. Many viruses including HCMV, Rubella virus, lymphocytic choriomeningitis virus, HIV, herpes simplex virus, measles virus, varicella zoster virus, mumps virus and West Nile virus have been documented as the causative agents for a variety of hearing loss. Among these viruses, congenital HCMV infection is recognized as the most common cause of SNHL in infants [8].

The pathogenesis of HCMV-related hearing loss is still not fully understood. Both direct virus-mediated cytopathology and virus-induced inner ear inflammation have been implicated in the auditory pathogenesis [6,9]. Viral antigens have been detected in the spiral ganglion, organ of Corti, scala media, and Reissner’s membrane in the terminated human fetuses infected by HCMV [10,11]. Similarly, in a murine model of murine CMV (MCMV) infection, MCMV encoded IE1 protein was detected in the spiral ganglion, spiral ligament, stria vascularis, and the bone marrow of the temporal bones, but MCMV infection did not lead to obvious anatomical destruction in the cochlea [10]. However, a recent study reveals stria dysfunction caused by MCMV infection and relates that to initial auditory threshold losses. MCMV could spread to the inner ear and damage the stria vascularis, which disrupts the endocochlear potential and consequently attenuates HC transduction [12]. Other evidence suggests that hearing loss is related to CMV induced immune response. When intracranially injected to neonatal mice, MCMV directly infected both cochlear perilymphatic epithelial cells and spiral ganglion neurons (SGNs) but not HCs. Interestingly, MCMV was cleared in the cochlea by 14 days post-infection (dpi), but HC loss occurred between 14 and 21 dpi. This postponed HC loss suggests that the inflammatory response might be responsible for the HC death [13,14,15,16,17], but the cell death pathways are still unclear. Hearing loss was further demonstrated to be correlated with the expression of MCMV-induced proinflammatory cytokines and chemokines, but not the viral load in the cochlea [8]. Though MCMV and HCMV are similar, there are still considerable differences between these viruses, particularly that MCMV cannot spread through the placenta like HCMV. Thus, any findings based on the mouse models should be cautiously interpreted, and innovative animal study designs are needed to improve modeling HCMV transmission and pathogenesis in vivo.

## 3. HCMV May Induce Hearing Loss through Connexin 43 Suppression

Many genes involved in genetic congenital hearing loss encode gap junction proteins, which belong to the connexins (Cx) gene family. Connexins are co-translationally inserted into the endoplasmic reticulum and transported through the Golgi and the trans-Golgi to the plasma membrane [18]. Connexins oligomerize into hexameric pores called connexons and assemble to form gap junction channels which physically connect two adjacent cells and enable direct cell-to-cell communication via the exchange of ions and small molecules [19]. Mutations in connexin genes—Cx26 (GJB2), Cx30 (GJB6), Cx29 (GJC3), Cx31 (GJB3) and Cx43 (GJA1)—have been identified as the culprits for hearing loss with distinct pathological changes in the cochlea [20]. Mutations in Cx43 are associated with non-syndromic autosomal recessive deafness and Cx43 was reported as the second most common mutated gene associated with SNHL in a cohort study in Taiwan [21,22].

The impacts of Cx43 mutations on hearing loss have been examined in a few Cx43 mutant mouse models. Cx43 G60S mutation led to ~80% gap junction channel function loss and the mutant mice displayed severe hearing loss. However, hearing loss was not observed in the mice carrying the Cx43 I130T mutation, which impaired ∼50% Cx43 function [23,24,25]. Similarly, 10-month-old Cx43+/− mice merely exhibited a modest decrease in hearing capability [25]. These findings further support that only severe loss of Cx43 function results in hearing loss in mice. Unexpectedly, loss of Cx43 in these mutant mice did not lead to mature HCs loss. In addition, Cx43 was implicated in cisplatin-induced hearing loss. In cisplatin-treated House Ear Institute-Organ of Corti 1 cells *in vitro*, Cx43 expression was reduced and its trafficking to cell membranes was interrupted, which resulted in altered gap junction communication and eventually cochlear cell death [26]. Taken together, not only loss-of-function mutations in Cx43 but also downregulation of Cx43 expression contributes to hearing loss.

Downregulation of Cx43 by HCMV infection has been reported in a variety of cell lines. In a screen of the cellular proteome in human fetal foreskin fibroblasts infected by HCMV, Cx43 expression was downregulated by infection with both HCMV strains Toledo and AD169. The Cx43 protein downregulation was apparent by 24 h post infection (hpi) with the protein levels dropping below detectability at 144 hpi [19]. It was later found in glioblastoma multiforme cells and fibroblasts that HCMV immediate early (IE) proteins participated in Cx43 downregulation by reducing its transcription and/or promoting Cx43 protein degradation through proteasome, which led to disruption of gap junction-mediated intercellular communication [27].

In rat cochlea, Cx43 expression was robust in peripheral neurite projections to HCs till the onset of hearing (postnatal day 17, P17) and Cx43 presence in the synaptic terminals decreased dramatically thereafter, implying a role of Cx43 in the cochlear synaptogenesis [28]. In MCMV-infected mice, these neurites connecting the cochlear HC and SGN nerve terminals were disrupted by MCMV induced inflammation in cochlear. This impairment could be mitigated by treating MCMV-infected mice with anti-inflammatory drug corticosteroid between P3–P20, which preserved HC-SGN synapse density and improved hearing [16,28]. It would be very interesting to see whether CMV infection or CMV-induced inflammation disrupts Cx43 expression in SGN neurites and how this interaction is involved in CMV-induced hearing loss in animal models. Also, further comprehensive studies are needed to examine the influences of HCMV infection on the expressions and functions of other connexin family members, which might gain new insights about the mechanism of HCMV-related hearing loss.

## 4. HCMV Regulates Wnt Signaling Pathway, An Essential Pathway in Inner Ear Development

Abnormal development of the inner ear leads to hearing loss: about 20–30% of children with SNHL have temporal bone abnormalities such as cochlear and vestibular abnormalities [29]. The inner ear development is delicately regulated by a few important signaling pathways. Exploring the roles of these pathways in auditory organ formation and their interplay with HCMV will help to understand the mechanisms of CMV-induced hearing loss.

The Wnt signaling pathway is a major driving force behind a number of key molecular events during embryonic development, such as cell fate decision and cell migration followed by tissue patterning [30]. The Wnt ligands are a family of secreted hydrophobic and cysteine-rich glycoproteins which bind to the extra-cellular cysteine-rich domain of Frizzled (Fzd) family receptors to initialize Wnt signaling. There are three types of receptor activation: the canonical Wnt/β-catenin cascade, the noncanonical planar cell polarity (PCP) pathway and the Wnt/Calcium pathway [30]. Among them, the canonical Wnt/β-catenin is the most important pathway involved in ear development and it is discussed in detail below. The PCP pathway also contributes to the staircase-like pattern formation of HC stereocilia and the polarized extension of the organ of Corti [31]. The participation of Wnt/Calcium pathway in inner ear development has not been reported.

The canonical Wnt pathway starts with Wnt ligands binding to Fzd receptors on the cell surface and ends with transcription of the target genes regulated by the T-cell/lymphoid enhancer-binding transcription factors (TCF/LEF), with β-catenin serving as a major second messenger [32]. β-Catenin was discovered as a subunit of the cadherin protein complex on the cell surface. Cytoplasmic β-catenin is constitutively degraded through proteasome by the β-catenin destruction complex, a large multiprotein assembly which comprises of β-catenin, two tumor suppressor proteins Axin and adenomatous polyposis coli (APC), casein kinase 1(CK1) and glycogen synthase kinase 3 (GSK3) [33]. Upon Wnt stimulation, Fzd receptors recruit Dishevelled (Dvl) protein to the plasma membrane where Dvl provides a docking site for axin and GSK3β and promotes LRP5/6 phosphorylation, resulting in the disassembly of the β-catenin destruction complex [34]. Released β-catenin translocates to the nucleus and interacts with TCF/LEF transcription factors to regulate the target gene expression (Figure 2) [35].

The Wnt/β-catenin pathway participates in a series of events during inner ear development, from the early otic vesicle formation to the late sensory epithelial cells fate decision (Table 1) [32]. β-Catenin is required for HC generation and patterning during cochlear development. Conditional knockout of β-catenin during sensory epithelium development inhibited the differentiation of sensory progenitors into HCs, while overexpression of β-catenin led to more immature HCs and abnormal expansion of the Organ of Corti [36]. In addition, differential expression of GSK3 along the radial axis of the cochlear spiral during cochlear development was observed, and inhibition of GSK3 increased the overall number of HCs with the IHCs number growing while the OHCs number shrinking [37]. However, this phenotype did not seem to be an outcome of the altered canonical Wnt signaling pathway [37].

HCMV targets the Wnt pathway and regulates its activity either positively or negatively in different scenarios. In colorectal cancer-derived stem cell-like cells, HCMV infection dramatically increased the gene expression of Wnt pathway components WNT11, frizzled-7, GSK3β, and β-catenin [42]. Wnt/β-catenin pathway was also highlighted in a transcriptional profiling analysis of HCMV-US28 expressing fibroblasts [43]. US28 is a chemokine receptor homolog encoded by HCMV. In a US28 transgenic mouse model, expression of US28 in intestinal epithelial cells significantly increased β-catenin protein level by inhibiting GSK-3β function and thus enhanced the expression of Wnt target genes (Table 2) [44]. However, the modulation of β-catenin by US28 is not through the classical Wnt signaling pathway, but depends on the Rho-Rho kinase (ROCK) pathway [45]. Other studies suggested that the Wnt pathway can be downregulated by HCMV. In human foreskin fibroblasts, HCMV infection altered subcellular distribution of β-catenin and resulted in its juxtanuclear accumulation with enhanced degradation, which reduced the transcriptional activities mediated by β-catenin (Figure 2) [30,46,47]. Moreover, HCMV reduced the poly-ADP-ribosylation activity of Tankyrase and consequently stabilized Axin1, a negative regulator of the Wnt pathway [47]. All the evidence suggests that HCMV can regulate Wnt pathway, however, how this interaction affects inner ear development is still unknown and more in vivo studies are needed to elucidate the underlying mechanism.

Interaction between Wnt/β-catenin signaling and Cx43 has also been reported. Cx43 was found to partially colocalize with β-catenin at the cell membrane and it translocated to the nucleus along with β-catenin upon Wnt pathway activation [51]. Cx43 was considered as a component of the β-catenin destruction complex because it interacts with β-catenin and CK1 of the destruction complex. Furthermore, knockdown of Cx43 resulted in β-catenin accumulation in the nucleus in the absence of Wnt activation and this Wnt signaling malfunction may contribute to Cx43-related hearing loss [51]. 

## 5. HCMV Perturbs Notch Signaling Pathway, Alters Cell Fate Decision, and Affects Inner Ear Development

As an evolutionarily highly conserved signaling pathway, the Notch pathway is critical for the development of most organ systems. It plays multiple roles during the inner ear development, from the origination of the otic placode alongside the central region of the embryonic hindbrain, to the mosaic cell pattern formation of HCs and SCs through lateral inhibition. The mammalian Notch pathway mainly consists of four transmembrane Notch receptors (Notch1–4) and five DSL (Delta/Serrate/Lag-2) ligands (Jagged 1/2 and Delta-like (Dll) 1/3/4). The transmembrane Notch receptor has an extracellular domain with multiple EGF-like repeats and a Notch intracellular domain (NICD). Binding of the DSL ligands to the Notch extracellular domain (NECD) triggers sequential proteolytic cleavages of Notch by the metalloproteases of the ADAM (A Disintegrin And Metalloprotease) family and γ-secretase. This process releases the NICD from the membrane, which translocates into the nucleus and forms a transcriptional complex with CSL (CBF1/SuH/Lag-1) to regulate the expression of Notch target genes such as the *Hes* (hairy enhancer of split) and *Hey* (hairy/enhancer of split-related with YRPW motif) families of basic helix–loop–helix transcription factors. Activation of Notch induces the expression of Sex-determining region Y (SRY)-box2 (SOX2), a core transcriptional factor for stem cell self-renewal and pluripotency; thus SOX2 has also been considered as a direct target of Notch pathway [52]. Notch effectors Hes1 and Hes5 facilitate the interaction between Janus kinase 2 (JAK2) and signal transducer and activator of transcription 3 (STAT3) which promotes STAT3 phosphorylation and translocation into the nucleus to drive SOX2 transcription (Figure 3) [53,54].

In the early stages of ear development, Notch pathway is involved in the otic placode induction, which later gives rise to the entire inner ear. The Notch1 receptor, its ligands (Jag1 and Dll1) and the downstream effector Hes1 are expressed in the otic placode, induced by Wnt signaling [38,55]. Notch signaling cooperates with Wnt signaling to refine the otic placode boundary [55]. Loss of Notch1 function diminishes Wnt signaling and leads to a reduction in the otic placode size, but the change is much smaller than that observed in conditional β-catenin–knockout mice in which Wnt signaling is blocked [56].

In addition, Notch signaling has been shown to regulate both the neurogenic process and the prosensory specification (Figure 4). During the transition of the otic placode into the otic vesicle, neuroblasts delaminate from the otic epithelium and form the cochleovestibular ganglia which later differentiate into the vestibular and auditory neurons to innervate the HCs. Neuroblast formation is regulated by Notch pathway through lateral inhibition. Notch1 is broadly expressed in the otic epithelium, but Notch ligand Dll1 and neurogenic transcription factor Neurogenin1 (Neurog1) are only expressed in the presumptive neuroblasts. These Dll1-expressing cells signal to neighboring cells by the Notch signaling pathway, which downregulates Neurog1 in those cells and thus prevents them from delaminating as neural precursors. Prosensory specification promotes some cells of the otic vesicle to form the prosensory cells which eventually give rise to the sensory organs of the inner ear containing mechanosensitive HCs. Two prosensory markers, SOX2 and the Notch ligand Jag1, are initially expressed broadly in the otic vesicle. Early SOX2-expressing cells include prospective prosensory cells but most of them will eventually lose SOX2 and develop into non-sensory tissues [57,58]. SOX2 is required in the differentiation of the prosensory cells to the HCs and SCs. Loss of SOX2 results in impaired development of the inner ear with fewer and disorganized HCs [38,58,59,60]. The SOX2 expression is positively regulated and maintained by Notch signaling to advance the formation of all inner ear sensory organs [38,60]. The Notch pathway regulates prosensory specification via lateral induction, a process by which Jag1-expressing cells induce nearby prosensory cells to upregulate Jag1 expression and therefore create a positive feedback loop so that all the cells in a cluster adapt a common prosensory cell fate (Figure 4). Jag1 is expressed in all prosensory domains and Jag1-mediated Notch signaling is essential for establishing the prosensory regions of the inner ear during early development [40].

Other Notch pathway components have also been reported to regulate cellular differentiation in the inner ear development. HCs express the Notch ligands Dll1, Dll3 and Jag2 and these ligands activate Notch signaling in neighboring cells to express genes such as Hes1, Hes5 and Hey1, which cooperate to induce the supporting cell fate [41]. The ligand Dll1 functions synergistically with Jag2 in regulating HC differentiation in the cochlear (Table 1) [39]. The expression of the Jag2 and Dll1 is essential for HC differentiation and is positively regulated by transcription factor Atoh1, which is in turn also negatively regulated by Notch signaling (Table 1) [61]. In addition, a few Notch effectors of the *Hes/Hey* family are expressed in the developing cochlea, where they function redundantly or cooperatively to ensure the proper cell alignment, polarity and cell numbers of HCs and SCs (Table 1) [41]. Overexpression of Hes1 or SOX2 prevents HC differentiation induced by Atoh1, suggesting the balance between Atoh1 and Hes1 or SOX2 is crucial for maintaining the appropriate number of HCs [59,62,63,64].

HCMV infection can interfere with organ development by altering cell fate decision through perturbing the Notch pathway. Proper differentiation of neural progenitor cells (NPCs) to mature neurons and glial cells is regulated by Notch signaling [65]. HCMV infects NPCs and causes premature and abnormal differentiation of NPCs [66]. The infection dysregulates Notch signaling by downregulating and altering the subcellular localization of the ligand Jag1 and effective receptor NICD1, and HCMV tegument protein pp71 has been identified to play an important role in this interaction (Figure 3 and Table 2) [50]. As a multifunctional protein, pp71 regulates viral gene expression, mediates host protein proteolysis, and contributes to the viral evasion of the immune response [67]. On the other hand, HCMV infection could upregulate the expression of NICD and Notch1, which increased the proliferation of U251 glioma cells [68]. As an important downstream effector of Notch signaling pathway, Hes1, with its oscillatory expression, plays an essential role in maintaining NPCs cell fate and fetal brain development. HCMV infection disrupts Hes1 rhythm and downregulates Hes1 expression. Loss of Hes1 suppressed NPCs proliferation and neurosphere formation, and resulted in NPCs abnormal differentiation [48,69]. Further study showed that HCMV immediate early protein IE1 downregulated the expression of Hes1 by promoting Hes1 ubiquitination and degradation through the proteasome [48]. IE1 is the first viral gene product newly synthesized upon infection to counteract intrinsic and innate immunity. It serves as a promiscuous transcription activator and interacts with STAT family members to attenuate the activation of IFN-stimulated genes [70]. Similarly, IE1 can reduce SOX2 expression by sequestrating its upstream regulator STAT3 in nuclei in an inactive unphosphorylated form (Figure 3 and Table 2) [49]. All these evidence regarding the interactions between HCMV and Notch signaling pathway reminds us that HCMV related hearing loss might be a result of perturbed Notch signaling pathway by HCMV. More in vivo examination of the impact of HCMV on Notch pathway and its relationship to hearing loss may reveal new mechanism on CMV induced SNHL.

## 6. Current Treatments and Future Perspectives

Hearing loss has a critical adverse impact on life quality, so it is important and urgent to develop effective therapies to reverse it. Antiviral therapies have been recommended for neonates with congenitally CMV infections [71]. Infants diagnosed with symptomatic congenitally CMV infection have improved hearing after valganciclovir therapy in their neonatal period [72]. Subsequent studies further demonstrated that this antiviral treatment also helped to prevent hearing loss in newborns with an asymptomatic congenital CMV infection [8,73,74]. However, valganciclovir only decreased viral load by 1.5 log in a small number of patients and the overall clinical benefit was modest [72]. Thus, new antiviral strategies are needed to formulate more effective treatments.

Considering the pivotal roles of Wnt and Notch signaling pathways in inner ear development and their intricate interactions with CMV, targeting these pathways may provide a new direction for therapeutic development. Activation of Wnt signaling pathway by a GSK3β inhibitor has been exploited to generate sensory HCs from cochlear SCs expressing Leucine-rich repeat containing G-protein-coupled receptor 5 [75]. In addition, Wnt pathway inhibitors have been shown to suppress HCMV replication and viral gene expression (IE2, UL44 and pp65) in human foreskin fibroblast [76]. It would be very encouraging if these compounds could attenuate CMV-related hearing loss in an animal model. Notch signaling blocker LY411575 (γ-secretase inhibitor) showed favorable effect on hearing loss in mice [41]. In mice with noise-induced cochlea damage, oral administration of LY411575 helped to regenerate OHCs and partially reversed hearing loss. Direct delivery of LY411575 to the inner ear at high dosages promoted SCs transdifferentiation into HCs in the mature cochlea. Thus, a two-step strategy was proposed as a better approach for long-term HCs regeneration [77,78]. Wnt/β-catenin pathway was first activated by an effective GSK3β inhibitor 6-Bromoindirubin-3′-oxime (BIO) in cochlear explant cultures; Notch pathway was then shut down with a γ-secretase inhibitor (DAPT) 3 days later. This combination treatment promoted the mitotic regeneration of HCs and increased the total HCs number in cultured neonatal cochlea with neomycin-induced damage. The advantage of this strategy over single pathway interference is that it induces more HCs regeneration and partially preserves the SCs number.

Gene replacement therapy is another innovative therapeutic approach for hearing loss and it has shown promising results in animal models. Conditional deletion of Cx26 in the inner ear of mouse resulted in HC loss and hearing impairment. Deliver a Cx26 replacement gene in cochlear organotypic cultures by ovine adeno-associated viral (AAV) vectors restored Cx26 protein expression and repaired gap junction coupling [79]. Further in vivo study in the Cx26 mutant mice demonstrated that AAV-mediated Cx26 expression reduced cell death in the organ of Corti and degeneration of spiral ganglion neurons in the cochlea [79,80]. Future investigation on the restoration of Cx function in the mutant mice of other Cx family members or congenital CMV-infected mice will shed light on the complicated roles of connexins in hearing loss due to genetic causes or CMV infection.

The detailed mechanism of CMV-induced hearing loss remains elusive. Previous studies were more focused on the direct cytotoxic effect and inflammatory response elicited by CMV. In this review, we highlight the significance of CMV in regulating Wnt and Notch signaling pathways during ear development. More research on these interactions in vivo will definitely yield profound insights about the mechanism of CMV-induced hearing loss, and reveal new targets for therapeutic interventions. 

## Figures and Tables

**Figure 1 viruses-13-00623-f001:**
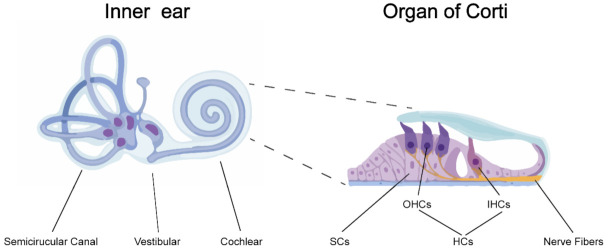
Anatomy of the inner ear and the organ of Corti. The inner ear is divided into three sections: the semicircular canals, the vestibular and the cochlear. The organ of Corti is located inside the cochlear duct, responsible for transducing sound vibrations into nerve impulses. The organ of Corti consists of three rows of outer hair cells (OHCs) and one row of inner hair cells (IHCs) interspaced by supporting cells (SCs).

**Figure 2 viruses-13-00623-f002:**
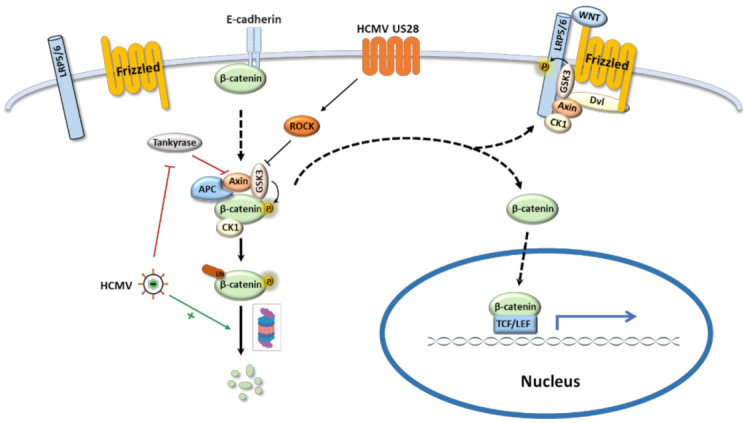
HCMV regulates the canonical Wnt/β-catenin signaling pathway. When Wnt pathway is off (Wnt receptor complex is not bound by a ligand), β-catenin is degraded through proteasome by the β-catenin destruction complex; the transcription of Wnt target genes is inhibited in the nucleus. When Wnt signal pathway is on and activated by binding Wnt ligand with the Frizzled receptor, the receptor recruits Dvl as a docking platform for components of β-catenin destruction complex, which promotes LRP5/6 phosphorylation and frees β-catenin. The freed β-catenin translocates to the nucleus where it interacts with TCF/LEF transcription factor to drive downstream target gene expression. HCMV regulates Wnt signaling pathway by the following routes: (1) HCMV US28 inhibits GSK3 activity via ROCK pathway and induces β-catenin translocation into the nucleus for Wnt target gene expression. (2) HCMV decreases poly-ADP-ribosylation activity of Tankyrase which stabilizes Axin and suppresses Wnt pathway. (3) HCMV induces β-catenin degradation and attenuates Wnt target gene expression.

**Figure 3 viruses-13-00623-f003:**
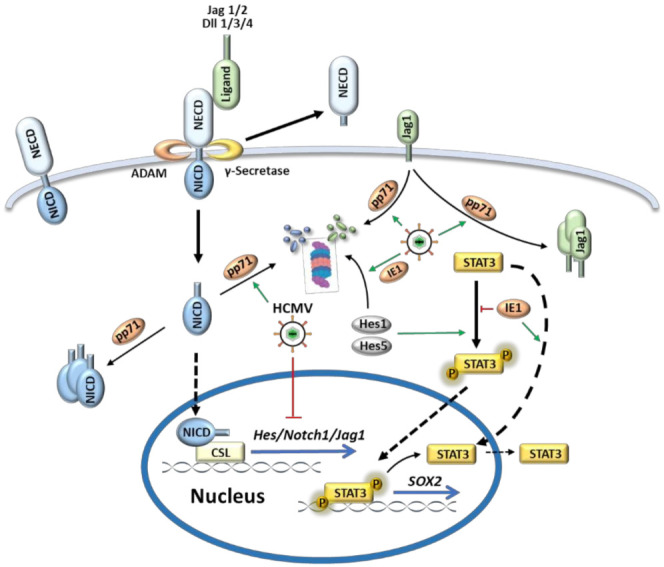
HCMV dysregulates the Notch signaling pathway. When Notch ligands (Dll1/3/4 and Jag1/2) bind with Notch receptors, proteolytic cleavage of Notch receptors by ADAM and γ-secretase complex releases Notch intracellular domain (NICD). Then NICD translocates into nucleus and binds with CSL (CBF1/SuH/Lag-1) to activate transcription of target effector genes (*Hes* and *Hey*). Hes1 and Hes5 can facilitate STAT3 phosphorylation which leads to STAT3 shuttling to nucleus for SOX2 transcription. HCMV tegument protein pp71 dysregulates Notch signaling by altering the subcellular localization of the ligand Jag1 and NICD1 and promoting their degradation through proteasome. In addition, two important genes (Hes1 and SOX2) in development of inner ear are regulated by HCMV IE1: IE1 promotes Hes1 degradation and sequesters unphosphorylated STAT3 in the nucleus, both of which contribute to decreased SOX2 transcription.

**Figure 4 viruses-13-00623-f004:**
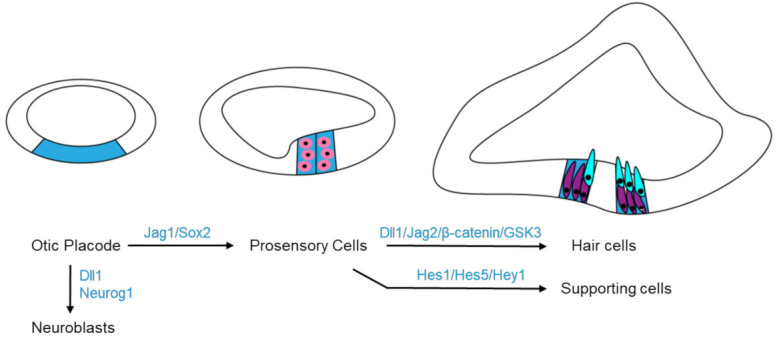
Molecules in the Wnt and Notch signaling pathways participate in cell fate determination during the inner ear development. Expressing Dll1 and Neurog1 on otic placode (blue) drives cell differentiating into neuroblasts, while Jag1 and Sox2 promote prosensory specification (pink). Later, Notch pathway (Dll1/Jag2) and Wnt pathway (β-catenin/GSK3) cooperate to drive hair cell (cyan) fate; but in the surrounding cells, Notch signaling targets Hes1, Hes5 and Hey1 to direct the supporting cell (purple) fate.

**Table 1 viruses-13-00623-t001:** Molecules of the Wnt and Notch signaling pathways involved in inner ear development.

Signaling Pathway	Proteins	Functions in Ear Development	Reference
Wnt	β-Catenin	HCs differentiation in cochlear	[36]
	GSK3	Development of the Corti	[37]
	Cx43	Development of neurite terminals between HCs and SGNs	[28]
Notch	Notch1	Determination of the otic placode’s size	[38]
	Jag1/Dll1	Establishment of the prosensory regions	[39,40]
	Jag2/Dll1	Differentiation of HCs in cochlear	[39]
	Hes/Hey	SCs differentiation in cochlear	[41]

**Table 2 viruses-13-00623-t002:** Viral proteins of HCMV involved in the Wnt and Notch signaling pathways.

Signaling Pathway	Proteins	Functions in Regulating Signaling Pathway	Reference
Wnt	US28	1. inhibiting GSK3 activity 2. inducing β-catenin translocation for Wnt target gene expression	[43,44,45]
Notch	IE1	1. promoting Hes1 degradation 2. sequestering unphosphorylated STAT3 in the nucleus	[48,49]
	pp71	1. altering the subcellular localization of the ligand Jag1 and NICD1 2. promoting degradation of Jag1 and NICD1 through proteasome	[50]

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
