# Peer review of "Hearing Loss Caused by HCMV Infection through Regulating the Wnt and Notch Signaling Pathways"

_viruses, 2021, doi:10.3390/v13040623_

Round 1

Reviewer 1 Report

In this article Huang and colleagues review the connection between fetal HCMV infection and hearing loss in newborn with a focus on the viral interference with the Wnt and Notch signaling during inner ear development. I think the article reviews an important and pressing public health concern and I found the manuscript very well written, concise, and easy to follow. I do not have any request for the authors, I recommend publication. 

Reviewer 2 Report

Manuscript from Huang, et al titled “Hearing loss caused by HCMV infection through regulating the Wnt and Notch signaling pathways” aims to introduce to the review reader “the impact of CMV infection on key players in inner ear development including the Wnt and Notch signaling pathways”. Authors have a reputable body of knowledge regarding HCMV infection and its effects in Wnt and Notch signaling. Authors did a very clear and thorough introduction into the rates of SNHL resulting from congenital HCMV infection and mentioning the lack of effective treatment or vaccine to prevent such outcomes.

Description of the anatomy of the inner ear and organ of Corti was very clear, a clarity further supplemented by the addition of Figure 1. Furthermore, authors did a commendable job bridging hearing loss-related HCMV pathogenesis considering the gaps in knowledge in this area—gaps that authors are working toward filling based on their publication record. Based on my review and minor comments, I recommend this review for publication in Vaccines.

Comments:

Overall, perhaps due to style, but please mind Oxford commas. The following are a few examples:

  • Line 47: comma between “…bifida” and “and”
  • Line 249, title: comma between “…decision” and “and”

Line 36: wording. From “…will develop SNHL subsequently.” to “…will subsequently develop SNHL.”

Lines 95-114: Well-written. For the sake of audience and review purposes, it would highlight the need to continue research in this area, as we are aware that there are limitations in the mouse models. As we know that MCMV and HCMV are similar, they are still considerably different and thus appreciating the limitations of this model.

Authors did an excellent job in highlighting the importance of connexin 43 via its mutations leading to loss of hearing. As this work mostly stems from mouse models, the previous comment regarding model limitations seems pertinent for the sake of this review.

Figure 2, Figure 3, and Table 1 were very clear.

Lines 336-356: For purposes of this review, a brief mention of best known/common function of IE1 or pp77. Perhaps the addition of a table such as Table 1 but for HCMV proteins putatively involved in perturbing the described signaling pathways. This table could serve as a complement to Figure 3.
